# Safety of Alternative Proteins: Technological, Environmental and Regulatory Aspects of Cultured Meat, Plant-Based Meat, Insect Protein and Single-Cell Protein

**DOI:** 10.3390/foods10061226

**Published:** 2021-05-28

**Authors:** Joshua Hadi, Gale Brightwell

**Affiliations:** 1AgResearch Ltd., Hopkirk Research Institute, Cnr University Ave and Library Road, Massey University, Palmerston North 4442, New Zealand; Joshua.Hadi@agresearch.co.nz; 2New Zealand Food Safety Science and Research Centre, Massey University Manawatu (Turitea), Tennent Drive, Palmerston North 4474, New Zealand

**Keywords:** alternative proteins, cultured meat, plant-based meat, edible insects, single-cell protein, novel food, environmental issues, food safety

## Abstract

Food security and environmental issues have become global crises that need transformative solutions. As livestock production is becoming less sustainable, alternative sources of proteins are urgently required. These include cultured meat, plant-based meat, insect protein and single-cell protein. Here, we describe the food safety aspects of these novel protein sources, in terms of their technological backgrounds, environmental impacts and the necessary regulatory framework for future mass-scale production. Briefly, cultured meat grown in fetal bovine serum-based media can be exposed to viruses or infectious prion, in addition to other safety risks associated with the use of genetic engineering. Plant-based meat may contain allergens, anti-nutrients and thermally induced carcinogens. Microbiological risks and allergens are the primary concerns associated with insect protein. Single-cell protein sources are divided into microalgae, fungi and bacteria, all of which have specific food safety risks that include toxins, allergens and high ribonucleic acid (RNA) contents. The environmental impacts of these alternative proteins can mainly be attributed to the production of growth substrates or during cultivation. Legislations related to novel food or genetic modification are the relevant regulatory framework to ensure the safety of alternative proteins. Lastly, additional studies on the food safety aspects of alternative proteins are urgently needed for providing relevant food governing authorities with sufficient data to oversee that the technological progress in this area is balanced with robust safety standards.

## 1. Introduction

Global population is projected to reach 9.8 billion in the year 2050 [1]. This population growth entails a projected livestock production of 455 million tons in 2050 [2], which is 40% higher than the reported number in 2019 [3]. Currently, livestock production contributes to 14.5% of anthropogenic greenhouse gas (GHG) emission [4]. In particular, livestock production releases methane and nitrous oxide gases, which have higher global warming potential than carbon dioxide [5]. To avoid potential catastrophic events, global temperature increase should be maintained within 1.5 °C of the pre-industrial levels [6]. Considerable requirements for water and land further contribute to the environmental footprints of livestock production [7,8].

Mitigation efforts include improved feeding practices for better forage digestibility, manure management and diversification of crop and animal varieties [9,10]. Nevertheless, climate issues are a matter of great urgency and more radical solutions may be necessary to ensure food availability in an environmentally sustainable manner. Thus, the concept of alternative protein arises as an attempt to substitute conventional meats with other protein sources that require less intensive production means.

Several examples of novel protein sources are cultured meat, plant-based meat, insect protein and single-cell protein, which have gained interests from researchers and the food industry in the past few years. To our knowledge, current research has focused on the technological and industrial application of these alternative proteins, but their safety aspects remain poorly described. In this review, we attempt to summarize their current food safety status to support future developments of safe alternative proteins, including brief descriptions of relevant technological backgrounds, environmental impacts and regulatory framework.

## 2. Cultured Meat

### 2.1. Cell Type and Culture Media

Cultured meat, also known as in vitro, lab-grown or cell-based meat, is derived from animal stem cells that are cultivated in controlled settings. Currently, the two main stem cells considered to be the most suitable for culturing meat are embryonic stems cells or satellite cells [11]. The main steps involved in the production of cultured meat include the isolation of stem cells from an animal biopsy, followed by the proliferation and differentiation of these isolated stem cells into desired tissues (for example, skeletal muscles) in a cell culture medium [11,12]. In the process, the growing cells can be attached to scaffolding materials, such as collagen-like gel polymers, which serve as a support network for the tissue development [11,12]—potential polymers to be used as scaffolds are listed elsewhere [13].

Key research milestones in the production of cultured meat include the patent for in vitro meat filed by van Ellen et al. in 1999 [14] and the report on in vitro cultivation of fish skeletal muscles in 2002 [15]. In 2013, the first cultured beef was produced by researchers at Maastricht University and was sampled in London [16]. However, as only satellite cells were used, the product only contained skeletal muscle fibers, but not other meat components, such as fat and connective tissues [16]. Thus, for cultured meat to emulate the complex structure of animal-based meats, additional cell sources have been proposed, including adipocyte tissue-derived stem cells and endothelial cells [12]. As reviewed by Ben-Arye and Levenberg, recapitulation of complex meat structures via tissue engineering needs to include skeletal muscles (myogenesis), extracellular matrix (fibrogenesis), microvascular networks (vascularization) and intramuscular fats (adipogenesis) [17].

Given the need for an array of cell types in the production of cultured meat, there is a practical advantage in utilizing pluripotent embryonic stem cells over their multipotent counterparts derived from adult animals (for example, satellite cells). In 2018, Bogliotti et al. reported on a stable culture of bovine embryonic stem cells [18]. However, ethical issues may arise from the use of embryonic stem cells, and thus there is still a need for alternative sources of stem cells. One of these sources is the induced pluripotent stem cells (iPSC) derived from adult cells, which were first discovered in 2006 by Takahashi and Yamada [19]. About a decade later, researchers generated in vitro skeletal muscles from porcine iPSC [20]. The main findings presented in these studies [18,19,20], along with those from other relevant publications, are summarized in Table 1.

Stem cells must be cultured in a suitable medium. Currently, animal serum-based media, particularly those containing fetal bovine serum (FBS), are the most commonly used [12]. In cell culture media, serum provides the growing cells with a range of essential nutrients, including hormonal and differentiation factors for cell proliferation (for example, growth hormone and insulin), transport proteins (for example, transferrin and transcortin) and growth factors (for example, epidermal, endothelial, fibroblast and insulin-like growth factors) [23]. However, for safety and ethical reasons, there have been attempts to develop serum-free media. While there are over 100 serum-free media formulations, growth requirements vary with cell type, and thus a universal serum-free medium may not be attainable [24].

Several serum-free media have been found to sustain the in vitro growth of bovine muscle stem cells. Kolkmann et al. found that three commercial serum-free media (FBM^TM^, TesR^TM^ and Essential 8^TM^) were able to promote the growth of primary bovine myoblasts, albeit the cell numbers after six days did not reach the level found in a standard growth medium (Dulbecco’s Modified Eagle’s Medium with 30% serum) [25]. Nevertheless, cost optimization will be needed for industrial-scale production of cultured meat, particularly regarding the high price of growth factors in serum-free media, which could account for up to 99% of the total cell culture medium cost (as estimated using ingredients of Essential 8^TM^ for a hypothetical 20,000 L batch) [26]. To alleviate these cost issues, a group of researchers has proposed substituting serum with yeast extracts or several hydrolysates derived from food by-products—these by-products were chicken carcass, cod backbone, pork plasma, eggshell membrane or egg white powder—all of which were found to sustain the proliferation of bovine skeletal muscle cells grown in serum-free media to a degree comparable to cells cultured in full-serum conditions [27]. Cyanobacterial hydrolysates have also been proposed as a nutrient source for culturing meat, albeit there is still a lack of experimental data, and thus this remains a subject of future studies [28].

### 2.2. Potential Food Safety Risks of Cultured Meat: Virus, Prion and Foreign Genes

Given that cultured meat is almost exclusively produced in a laboratory, it can be considered to be less susceptible to zoonotic diseases than conventional meat products. However, there are gaps in the current knowledge of food safety related to cultured meat, especially due to the fact that the majority of research efforts are focused on the optimization of production means. The most apparent safety issues may arise from the use of animal serum in the culture medium. As outlined by the European Medicines Agency [29] and the United States Department of Agriculture [30], all bovine-derived serum should be free of the following viruses: bovine viral diarrhea virus, reovirus 3, rabies virus, bluetongue virus, bovine adenovirus, bovine parvovirus and bovine respiratory syncytial virus, regardless of their geographical origin [31].

Another potential hazard in animal serum is the pathogenic and infectious prion (PrP^SC^), particularly due to possible cross-species and blood-related transmissions. PrP^SC^ is a misfolded prion protein commonly associated with transmissible spongiform encephalopathies, such as scrapie in sheep, chronic wasting disease in deer, bovine spongiform encephalopathy (BSE) in cattle and variant Creutzfeldt-Jakob disease (vCJD) in humans [32]. As an acquired form of Creutzfeldt-Jakob disease—the other forms being sporadic and genetic—vCJD has been associated with the consumption of food products derived from infected cattle. The most prominent example is the disease onset of vCJD in the United Kingdom that peaked in the year 2000, which was about 6–7 years after the peak of the BSE epidemic in the same region [33]. Another study confirmed that cross-species transmission of PrP^SC^ from BSE-infected cattle to humans was possible, using transgenic mice as a model [34]. Others have also demonstrated that blood components may act as vectors for prion disease transmission [35,36].

In addition, the production of cultured cells for human consumption is unprecedented and would require further assessments, particularly regarding the use of genetic engineering. For example, the generation of iPSCs involves the introduction of exogenous genes (i.e., transcription factors) into the genome of somatic cells (Table 1). Genetic engineering may also be used to improve the means of production—for example, by increasing cell culture density through the inhibition of Hippo signaling pathway, as demonstrated in a patent application submitted by researchers at Memphis Meat [37].

The issues mentioned in this section are still mostly theoretical, as to our knowledge, the current literature lacks studies that directly assess the food safety risks related to cultured meat. Future research should aim at identifying the safety issues of cultured meat, particularly within the context of mass-scale production. These may include the retention and infectivity of viruses or PrP^SC^ within the final structure of cultured meat and also the health implications, if any, of ingesting meat analogues derived from genetically modified cultured cells.

## 3. Plant-Based Meat

### 3.1. Ingredients and Processing Technologies

The early use of plant-based ingredients, such as fermented soybean cake (i.e., tofu and tempeh) or wheat (i.e., seitan), as meat analogues could be traced back to the Asian communities in the 10th century [38,39]. However, these products are not able to emulate the sensorial properties of animal-based meat. Instead, texturized vegetable proteins (TVP) can be used as a potential replacement of conventional meat and are commonly derived from soy proteins [40], or to a lesser extent, from wheat glutens [41,42] and legume proteins (for example, pea and chickpea) [43,44]. In association with the Institutes of Food Technologists, Egbert and Borders proposed a composition of plant-based meat (Table 2), which is consistent with the recipes used by several existing animal-free meat companies, such as Impossible Food, Beyond Meat and Gardein [45,46].

Currently, plant-based meats are mainly produced through thermoplastic extrusion [42,43,48,49,50]. This process can be categorized based upon the amount of water added, i.e., low moisture (20–35%) or high moisture (50–70%) [51]. Both product types are made in three main steps: (1) pre-conditioning of the raw materials outside of the extruder; (2) heating and compression inside of the extruder; (3) cooling of the die and processing of the final product (for example, cutting to desired pieces) [52]. In low-moisture extrusion, unhydrated protein concentrates may be directly introduced into the extruder, which would result in a final product that has a high water-binding capacity, and thus can be used as a meat extender (for example, in sausages or beef patties). On the contrary, high-moisture extrusion involves pre-extrusion hydration of the protein concentrates to a moisture content of 60–70%, followed by heating in the extruder that induces the formation of viscoelastic mass, and subsequently slow cooling to prevent the disruption of the newly formed product due to excessive material expansion [45].

The inherent differences between plant and animal proteins present a challenge to the production of plant-based meat products. As highlighted in one study, there were differences in the physical and nutritional features between TVP and animal-based meats, including their psychochemical properties, textural characteristics and amino acid retention after thermal processing [53]. This concern may be partially addressed by the use of potassium carbonate [50] and alginate [48] to achieve sensorial properties of plant-based meat that are comparable to cooked ground beef. Essential nutrients that are often missing from a vegetarian diet, such as vitamins B-12 and D, calcium, zinc, iron and long-chained n-3 (omega-3) fatty acids, could also be added post-extrusion to increase the nutritional value of plant-based meat [49]. However, future research is needed to further optimize the processing conditions of plant-based meat, for example, through the use of computational techniques to predict the sensorial properties of meat analogues as a function of operating conditions. A group of researchers used a genetic algorithm to optimize the texture and appearance of a meat analogue, i.e., extrusion process conditions that achieved the highest water and oil binding capacity and that also achieved product brightness [54].

Shear cell technology can be used to structure proteins through the combination of simple shear and elevated temperature, either in a conical shear cell device or in a cylindrical Couette cell [45]. The final structure depends on the process temperature, concentration of dry matter content (soy protein concentrates) and presence of polysaccharide (for example, soy fiber) [55]. Consistently, varying process conditions have been used for other types of protein or protein–polysaccharide combinations, such as soy protein isolates (SPI) [55,56], SPI and wheat gluten [57] or SPI and pectin [56,58]. Krintiras et al. have also demonstrated that scale-up is possible from a small Couette cell (200 g of sample) [59] to a larger one (approximately 7.5 kg of sample) [60], with the authors identifying no barriers to further upscaling [60].

Other available technologies include electrospinning [61] and 3D printing [62], although the feasibility of these techniques for the production of plant-based meat remains a subject of future studies.

### 3.2. Food Safety Risks of Plant-Based Meat: Bacteria, Anti-Nutrients, Allergens, Thermally Induced Carcinogens and Genetically Modified Soybean

Plant-based meat can carry pathogenic bacteria originating from the raw ingredients. Although most of these bacteria could be inactivated by the heat produced during the extrusion process, some endospore-forming bacteria, such as *Clostridium* spp. or *Bacillus* spp., may survive the heating regime [63]. In a research commissioned by the European Union (i.e., LikeMeat project), *Clostridium sporogenes* spores (ATCC 19404) in protein ingredients were rendered inactive (below detection level) by extrusion, albeit *Bacillus amyloliquefaciens* (AB255669 and LMA008A; <1000 colony forming units/g) were detected in the final protein extrudates, possibly due to re-contamination post-extrusion [63,64]. This notion of re-contamination was further enhanced by the identification of *Enterococcus durans*, *Exigobacterium acetylicum*, *Acinetobacter* spp. and *Staphylococcus* spp. in uninoculated samples [64]. Consistently, another study identified several bacterial contaminants in plant-based meat, which predominantly consisted of *Lactobacillus sakei*, *Enterococcus faecium* and *Carnobacterium divergens* [65].

Health risk of plant-based meat may also arise from the presence of anti-nutrients. Legumes have been associated with anti-nutrients, such as protease inhibitors, α-amylase inhibitors, lectins (phytohemagglutinin), polyphenols (particularly tannins) and phytic acid [66]. While these anti-nutrients can have health benefits, they have also been associated with negative physiological effects, including altered gut function (for example, inactivation of digestive enzymes or reduced iron absorption) and endocrine disruption, among others [66,67]. Nevertheless, as reviewed by Petroski and Minich, plant-based anti-nutritional compounds (lectins, oxalate, phytate, goitrogens, phytoestrogens and tannins) are mainly harmful when ingested in high quantities or in isolation, and they could be inactivated through cooking [67]. Others also found that extrusion at 170–180 °C reduced the amount of trypsin inhibitor, phytic acid and tannins in a meal derived from maize, soybean concentrate and cassava starch [68].

It is well-established that legumes contain allergens that cause mild to life-threatening conditions. The majority of these allergens belong to three protein families—namely, storage proteins (with two main superfamilies prolamins and cupins), pathogenesis-related proteins (for example, Gly m 4 in soybean) or prolifins (for example, Gly m 3 in soybean). Legume-related allergens elicit IgE-mediated immunological reactions, with clinical manifestation primarily grouped into four categories: cutaneous (skin), gastrointestinal, cardiovascular and respiratory. Detailed discussions on legume allergies are provided elsewhere [69]. Given the uncertainty on whether thermal processing can reduce the allerginicity of legume proteins, particularly those derived from soybeans and peas, additional studies are warranted [70,71]. Several research reports suggest that treatments with high pressure, ultrasound and pulsed light can reduce the activity of allergens in soybean [72]. Nonetheless, clinical studies are also necessary to determine whether the reduced activities of legume allergens post-processing lead to significant physiological and immunological effects in live hosts, including humans.

While thermal processing is essential for reducing the activities of microorganisms, anti-nutrients and allergens, it may also induce the formation of carcinogens—particularly in processed meat products—for example, polycyclic aromatic hydrocarbons [73,74], nitrosamines [75,76] or heterocyclic aromatic amines [77,78]. In their review article, He et al. mentioned unpublished data on the detection of *N*-nitrosodiethylamine (15 µg/kg) in one sample of cooked plant-based meat [79]. Thus, more studies are required to determine the potential safety risk of these chemicals in plant-based meat [79].

As of 2017, the global adoption of genetically modified soybeans (also known as biotech soybean or GM soybean) reached 77%, which included glyphosate-tolerant variants and were primarily planted in Brazil, Argentina and the United States of America (USA) [80]. Concerns arise from the potential accumulation of glyphosate residues in these GM soybeans—or their associated food products—and the subsequent adverse effects upon ingestion [81]. These health concerns have been highlighted in studies using non-human animal models, such as Japanese quails [82] and crustacean *Daphnia magna* [83,84]. Current literature contains no data on the accumulation of glyphosate in plant-based meat.

Another GM ingredient in plant-based meat is soy leghemoglobin, derived from yeast (*Pichia pastoris*) expressing the leghemoglobin c2 gene from soybean (*Glycine max*). Jin et al. reported no potential allergenicity and toxigenicity associated with the use of recombinant soy leghemoglobin in food, which was a conclusion determined through a literature search, bioinformatics analyses on the amino acid sequence of leghemoglobin and 17 *Pichia* host proteins, and also in vitro pepsin digestibility of leghemoglobin [85]. Another study by Fraser et al. demonstrated no genotoxic effects of recombinant soy leghemoglobin on *Salmonella enterica* subsp. *enterica* serovar Typhimurium and *E. coli* (Ames test) as well as no adverse events in Sprague Dawley rats fed with the protein for 28 days at 750 mg/kg/day [86].

## 4. Insect Protein

### 4.1. Insect Species, Farming and Processing

Insects have been a part of the human diet for centuries, particularly in Asia and Africa [87]. According to the Food and Agriculture Organization of the United Nations (FAO), there are over 1900 insect species consumed around the world (Table 3) [88]. This practice of eating insects, also known as entomophagy, is sustainable due to the high amounts of protein and polyunsaturated fatty acid contained in edible insects, although there are variations across species [89,90,91,92]. Insects are also more effective in converting feed into edible body mass than farm animals [93]. These have made them an attractive option for expanded production to improve global food security.

Most edible insects are harvested from the wild, but they can also be semi-domesticated through habitat manipulation or reared in farms for a mass-scale production [88,94]. Wild harvesting and semi-domestication of insects are discussed in detail elsewhere [94], whereas our review article focuses on farmed insects. For example, edible palm weevil (*Rhynchophorus ferrugineus*) and crickets (*Acheta domesticus*, *Gryllus bimaculatus*, *Teleogryllus testaceus* and *Teleogryllus occipitalis*) are produced by local farmers in Thailand [95]. These farmers use containers (plastic drawers, concrete block pens, plywood boxes, etc.) to house the insects and nourish them with chicken feed (crickets) or plant-based feeds (for example, sago palm trunks for palm weevil or cassava leaves for crickets) [95].

Similar to other animals, insects require macronutrients (lipids, proteins and carbohydrates) and micronutrients (essential sterols and vitamins), which can be derived from animals, plants and yeast [96]. In particular, polyunsaturated acids, essential amino acids and sterols must be supplied in the feeds, given that insects lack the ability to synthesize these compounds in sufficient amounts [97]. As previously reported, the nutritional composition of feeds could influence the growth performance of mealworms (*Tenebrio molitor*, *Zophobas atratus* or *Alphitobius diaperinus*), including their fatty acid profiles, lipid contents, larval development time and progeny production [98,99,100]. As summarized by Morales-Ramos et al., strategies to optimize artificial dietary supplements for farmed insects include the determination of basic nutrient ratios (lipids, carbohydrates and proteins), feeding adaptations (for example, encapsulation of liquid nutrients) and feeding refinement (for example, feeding stimulants or nutritional adjustments for different growth stages) [97]. While organic wastes can be used as feeds for farmed insects, there are concerns about the potential transmission of pathogens from these waste materials, the lack of adequate nutrients present and the uncertain supply of organic wastes, particularly for a mass-scale production [88]. A review article by Varelas provides extensive discussion on the mass production of edible insects using food wastes as feeding substrates, including coverage of topics on the composition of different substrates, fermentation processes and characteristics of different insects fed with varying food wastes [101].

In addition to adequate nourishments, rearing conditions (for example, temperature, humidity and population density) need to be optimized [102]. Currently, there is a lack of data on the best method for processing and storing edible insects in a mass-scale production, particularly given that nutritional needs and rearing conditions vary across species [102]. Viral infections are another challenge for insect farmers, and this topic is comprehensively assessed in a review article by Maciel-Vergara and Ros, which includes discussion on the factors affecting the emergence of these pathogens in mass-scale rearing systems and also the measures to prevent or manage infections that range from simple sanitation interventions to advanced antiviral methods (for example, RNA interference and transgenic technologies) [103].

Post-harvest processing of edible insects traditionally involves degutting and thermal processes, such as boiling, frying, toasting, smoking, roasting and drying, which are particularly important for eliminating microbial contaminants and increasing the shelf-life of the final products [104]. More recently, other technologies have been used for the extraction of substances from edible insects, including ultrasound, enzymatic hydrolysis, supercritical carbon dioxide, sonication, soxhlet extraction and folch extraction [105].

### 4.2. Food Safety Risks of Edible Insects: Bacteria, Mycotoxins, Parasites, Allergens, Heavy Metals and Anti-Nutrients

Microbial contents of edible insects are affected by growth conditions and species. Metagenomic analyses revealed that microbial communities in yellow mealworm larvae (*T. molitor*) [106,107] and grasshoppers (*Locusta migratoria*) [107] were dominated by bacteria belonging to the phyla Proteobacteria and Firmicutes, although the two insect species contained distinct bacterial genera. Further, the abundance of Actinobacteria and Bacterioidetes differed considerably between the two studies [106,107]. Others have also reported variations in the abundance and the type of bacteria present in edible insects supplied by different companies [108,109]. These discrepancies may be attributed to differing farming practices between companies, as highlighted by Li et al. that gut microbiota of *T. molitor* larvae varied with rearing conditions (closed or open environment) [110].

Safety concerns can arise from the presence of pathogenic microorganisms in edible insects. Several bacterial genera, including *Cronobacter*, *Bacillus*, *Staphylococcus, Clostridium* and *Haemophilus*, have been identified in *T. molitor* or *L. migratoria* [106,107]. In another study, *Bacillus cereus* and several potentially pathogenic bacterial genera (for example, *Salmonella, Vibrio* and *Acinetobacter*) were detected in products derived from crickets (*A. domesticus*), locusts (*L. migratoria*) and yellow mealworm larvae (*T. molitor*) [108]. Bacterial endospores of *B. cereus* were also isolated from *T. molitor* and *A. domesticus*, with most strains (65%) containing cereulide plasmid, which may enable the production of a heat-resistant toxin [111]. Others found that non-pathogenic Enterobacteriaceae dominated in *T. molitor* [109,112,113]. Antimicrobial-resistance genes have also been reported in edible insects, which included *L. migratoria*, *A. domesticus* and *T. molitor* [114,115,116].

The presence of fungi, such as those belonging to the genera *Aspergillus*, *Penicillium* and *Fusarium*, in edible insects is a concern due to potential health risks associated with mycotoxins [117,118]. As reported by Musundire et al., aflatoxin B_1_ was present in edible stink bugs (*Encosternum delegorguei*), although the presence of the toxin depended on the containers used to store the insects [119]. Interestingly, contamination of feeding substrates with mycotoxins (for example, aflatoxin B_1_, deoxynivalenol, ochratoxin or zearalenone) did not seem to result in the accumulation of these toxins in several edible insect species, which was potentially due to the ability of these insects to metabolize mycotoxins [120,121,122,123]. Another study showed that *T. molitor* could resist several mycotoxins introduced to their feeds, albeit this depended on the fungal species from which the mycotoxins were derived [124]. These findings indicate that the digestion of mycotoxins may be specific to the insect species and the type of mycotoxin, and thus future research is required, including studies on the by-products of mycotoxin digestion by insects.

In addition to bacteria and fungi, parasitic pathogens may be transmitted via edible insects. For example, potential human parasites have been identified in 91 insect farms across six European countries, including *Cryptosporidium* spp., *Isospora* spp., *Balantidium* spp. and *Entamoeba* spp. [125]. Diseases have been associated with the ingestion of infected insects, as previously recorded for the transmission of *Trypanosoma cruzi* [126] and *Gongylonema pulchrum* [127].

Safety risk of food-borne viral pathogens in edible insects is low, as one study reported the absence of hepatitis A virus, hepatitis E virus and norovirus (genogroup II) in raw yellow mealworms (*T. molitor*) and crickets (*A. domesticus*) [111]. A similar conclusion was proposed in a report by the European Food Safety Authority (EFSA), in which a panel of experts declared that viral infections in edible insects were specific for insects, and thus not considered a hazard for humans [128].

These microbiological concerns can be mitigated by thermal interventions, such as blanching and drying [113]. However, Klunder et al. found that heating was only effective against Enterobacteriaceae in *T. molitor* and *A. domesticus*, but not against bacterial spores [129]. Instead, the authors proposed drying or acidification, including the use of lactic acid fermentation to preserve composite meals of sorghum and *T. molitor* [129]. Interestingly, Rumpold et al. found that indirect cold plasma treatment was effective in reducing the microbial load on the surface of *T. molitor*, albeit additional thermal (90 °C) and high pressure (600 MPa) treatments were required to achieve inactivation of the gut microbiota [130].

Similar to plant-based meat, edible insects may be a source of allergens. These include arginine kinase and tropomyosin, which are pan-allergens dominant in arthropods, as evident from reports on IgE cross-reactivity between edible insects and house dust mite or crustaceans [131]. However, the clinical significance of these allergens remains to be established [131]. The effects of processing on the allergenicity of edible insects are also unclear. In one study, enzymatic hydrolysis and thermal treatment eliminated cross-reactivity and allerginicity of insect extracts (*L. migratoria*), as tested by immunoblots and skin prick test [132]. Similarly, heat treatment reduced but did not eliminate the allergenicity of mealworms (*T. molitor*, *Zophobas atratus* and *Alphitobius diaperinus*) in samples taken from patients allergic to house dust mites and crustaceans [133]. Others demonstrated that thermal processing of *T. molitor* did not change its IgE binding in a basophil activation test nor in the skin reaction in a skin prick test (crustacean-allergic patients were used), although the solubility of several proteins was altered [134]. Consistent with the data that we collated in this review, conflicting results were also reported in a review article by Gier and Verhoeckx [135].

Heavy metals can accumulate in edible insects, which is dependent on the growth environments (wild or reared). Vijver et al. reported that cadmium, copper, lead and zinc present in soils (naturally or artificially contaminated) were retained in the body of *T. molitor*, with the extent of accumulation depending on the type of soil and metal [136]. Greenfield, Akala and van der Bank detected high concentrations of heavy metals in Mopane worms (*Imbrasia belina*) taken from two sites at a South African national park: cadmium, copper and zinc concentrations were respectively 15–21, 2–2.5 and 0.4 times higher than the recommended legal levels in the United Kingdom and European Union [137]. The concentration of manganese in *I. belina* was 20–67 times higher than the food safety standard set by the United States Food and Drug Administration [137]. Others found that contaminated feeds led to the accumulation of cadmium, lead or arsenic in soldier fly larvae (*Hermetia illucens*) and yellow mealworms (*T. molitor*) [123,138], with one study also suggesting that the accumulation rate was dependent on the insect species and type of metal present [138]. On the contrary, another group of researchers demonstrated an insignificant presence of arsenic, lead, chromium and mercury in reared grasshoppers (*Oxya chinensis* subsp. *formosana*) [139].

Available data suggest that there is a minimal food risk associated with anti-nutritional factors in edible insects. In India, a study found low levels of phenols and tannins (below 0.52%) in aquatic edible insects—namely, *Lethocerus indicus*, *Laccotrephes maculatus*, *Hydrophilus olivaceous*, *Cybister tripunctatus* and *Crocothemes servillia* [140]. Another study conducted in Nigeria reported that the larvae of *Cirina forda* contained acceptable levels of oxalate (4.11 mg/100 g) and phytic acid (1.02 mg/100 g) [141]. Similarly, EFSA declared that the amounts of oxalic acid, phytic acid, hydrogen cyanide and polyphenols in *T. molitor* were comparable to other foodstuffs [142]. However, researchers in Japan detected heat-resistant thiaminase, which is a risk factor for vitamin B_1_ deficiency, in the pupae of edible African silkworm (*Anaphe* spp.) [143]. Future investigations are required for other insect species.

## 5. Single-Cell Protein: Microalgae, Fungi and Bacteria

Single-cell proteins, also known as microbial proteins, are commonly derived from microalgae, fungi or bacteria. In their review article, Ritala et al. summarized available studies on potential fungal, microalgal and bacterial species for application in the production of single-cell proteins, including patents from the years 2001 to 2016 [144]. However, the safety aspects of this food category are still unknown, and thus this section was aimed at identifying potential safety hazards associated with single-cell proteins, including allergens, toxins and heavy metals.

### 5.1. Microalgal Protein: Species, Processing Technology and Food Safety Risks

Several of the most biotechnologically relevant microalgae include green algae (*Chlorella vulgaris*), *Haemotococcus pluvialis*, *Dunaliella salina* and spirulina (*Arthrospira maxima* or *Arthrospira platensis*), as shown in Figure 1 [145]—while spirulina is commonly categorized as a product derived from microalgae (blue/green), it is biologically classified as belonging to the phylum Cyanobacteria. As demonstrated in three studies, microalgae (*A. platensis*, *C. vulgaris*, *Chlorella pyrenodiosa, Isochrisis galbana* or *Tetraselmis* spp.) contained high levels of protein (w_protein_/w_dry mass_), albeit there were variations across species ranging from 27% in *I. galbana* to 64% in *A. platensis* [146,147,148]. Similarly, the contents of polyunsaturated fatty acids, carbohydrates and mineral elements (for example, calcium and potassium) vary with different microalgal species [146,147]. These variations in the nutrient contents of microalgae have also been reported in previous review articles [145,149,150].

Most commercial production of microalgae utilizes open-air systems, which can be divided into four categories: big ponds, tanks, circular ponds and raceway ponds. However, due to the disadvantages of these open systems, such as low productivity, contamination risk, difficulty with biomass recovery and problems of temperature control, closed systems have been developed, including tubular, flat panels and others. Microalgal biomass can be recovered by sedimentation, filtration, centrifugation or flotation. Subsequently, the biomass can be used as a whole, for example, after being spray-dried [149]. Alternatively, microalgal proteins can be extracted in two steps: (1) disrupting the cells through mechanical (for example, bead mill, homogenizer or ultrasonication) or non-mechanical (microwave, pulsed electric field, enzymatic, ionic liquid or chemicals) means; (2) separation of protein phase from other cellular debris by means of centrifugation, filtration or ultrafiltration, in combination with improving the dispersibility of protein in the aqueous solution [151]. A new technology of a three-phase partitioning system has also been proposed to extract proteins from *C. pyrenoidosa*, in which the microalgal components are divided into non-polar (upper), protein (middle) and polar (lower) phases [148].

Despite their nutritional values, microalgae may also carry harmful substances, such as heavy metals and toxins, particularly due to contaminations. Rzymski et al. found that certain food supplements derived from *Spirulina* spp. and *Chlorella* spp. contained cadmium, mercury and lead, albeit at levels below the provisional tolerable weekly intake, and this was likely due to the lack of quality control measures [153]. The contamination risk is particularly high when wastewater is used as a growing substrate, given that microalgae can uptake these heavy metals from the environment, either passively via surface sorption (living or non-living cells) or through metabolic-dependent activities (for example, active transport) [154,155]. Further, the rate of heavy metal uptake varies across microalgal species; for example, cadmium accumulation ranged from 0.02 mg_Cd_/g_dry biomass_ in *I. galbana* and 8–357 mg_Cd_/g_dry biomass_ in *A. platensis* to 1055.27 mg_Cd_/g_dry biomass_ in *Chaetoceros calcitrans* [155].

Toxin can accumulate in microalgae, particularly when the environment is contaminated with toxin-producing cyanobacteria, such as *Microcystis aeruginosa*. Two studies reported the contamination of microalgal dietary supplements (*Aphanizomenon flos-aquae*) with cyanobacterial hepatoxins—namely, microcystins, as confirmed by toxin detection (cPPIA, Adda-ELISA or LC/MS assays) and also the presence of microcystin-producing genes *mcyE* or *mcyB* (PCR analysis) [156,157]. Indeed, *A. flos-aquae* has the capability of producing cyanotoxins, such as anatoxin-a, cylindrospermopsin, microcystins and saxitoxins, and thus the presence of these cyanotoxins in *A. flos-aquae* products may be the result of direct toxin production by *A. flos-aquae* or cross-contamination with other toxin-producing cyanobacteria [158]. Along with microcystins, anatoxin-a and β-methylamino-L-alanine were detected in eight food products derived from *Arthrospira* or *A. flos-aquae* [159]. Interestingly, it has been reported that while toxins (microcystins or polymethoxy-1-alkenes) were absent in several dietary supplements derived from *Arthrospira* spp. and *Chlorella* spp., these products still exhibited cytotoxicity in human A549 cells [156] and adult zebrafish (*Danio rerio*) [160], although the reason for this phenomenon was unknown [156,160].

Several authors have reported allergic reactions to microalgae, including anaphylaxis after the consumption of spirulina-derived products (*A. platensis*) [161,162] or acute tubulointerstitial nephritis following the ingestion of *Chlorella* tablets [163]. Another case report detailed severe allergic reactions in a girl after swimming in a lake with blooming freshwater cynobacteria, and subsequent immunological tests showed IgE cross-reactivity with extracts of several cyanobacterial species—namely, *M. aeruginosa*, *Synechocytis* spp., *Synechoccus* spp., *Pseudanabaena* spp., *Oscillatoria* spp., *Lyngbya* spp. and *Arthrospira* spp. [164]. The β-chain of the C-phycocyanin protein has been identified as the main allergen in cyanobacteria [161,165], which may act alone or in a complex with other phycobiliproteins [166]. One study also reported that other unidentified proteins could potentially be allergens in several freshwater, marine and terrestrial cyanobacterial species [165]. Allergens in non-cyanobacterial microalgae are still poorly understood.

Microbial contamination poses an additional safety issue associated with microalgal food products. In one study, filamentous cyanobacterial contamination was detected in commercial *Chlorella* tablets, along with several non-pathogenic bacteria [167]. Further, genomic analyses have revealed the presence of several potentially pathogenic bacterial genera in spirulina products (*A. platensis*)—namely, *Pseudomonas*, *Flavobacterium*, *Vibrio*, *Aeromonas*, *Clostridium*, *Bacillus*, *Fusobacterium* and *Enterococcus* [168]. Another study reported the detection of *Clostridium* endospores in commercial *A. platensis* products, including toxin-producing and β-hemolytic isolates [169].

In addition to bacterial contamination, concerns about viral pathogens in microalgae have been raised due to the isolation of Acanthocytis turfacea chlorella virus 1 (ATCV-1), which is a chlorovirus commonly infecting green algae, in human oropharyngeal samples [170]. Subsequent analyses indicated that ATCV-1 resulted in decreased cognitive functions in humans and mice [170], potentially due to fact that this virus persisted and induced inflammatory factors in the macrophages [171]. The relevance of these findings to the food industry needs to be ascertained in future studies.

As described above, the processing of microalgal biomass involves minimal heat treatments. Thus, the removal of heavy metals, toxins, allergens and microorganisms from microalgae can be challenging. Available methods include heavy metal-binding ligand for removing heavy metals [172] or cold plasma for inactivating microorganisms [173]. Given that phycocyanin is a functional compound, it can be extracted from the cells of cyanobacterial microalgae (for example, through a high-pressure extraction process) and used as a separate product [174].

Pre-harvest preventative measures can also be taken with the use of microcystinase A enzyme (MlrA) to reduce the prevalence of *M. aeruginosa* in the environment and its concomitant production of microcystins. As reported by Liu et al., MlrA decreased the cell viability of *M. aeruginosa* through the degradation of extracellular microcystin-LR, inhibition of genes responsible for intracellular microcystin production (*mycA, mycB, mycD* and *mycG*) and impairment of photosynthetic ability, including lowered expression of photosynthetic genes *psbB*, *psbD*, *rbcL* and *fbp* [175]. The authors also found that the activity of MlrA was selective, as it did not affect *Synechocystis* sp. PCC 6803, which is a microcystin-nonproducing species [175].

### 5.2. Fungal Protein (Mycoprotein): Quorn^TM^, Potential Allergens and Mycotoxins

In the late 1960s, an effort was conducted to search for alternative proteins from starch-fermenting fungi, which led to the discovery of a filamentous fungus called *Fusarium graminearum* A3/5 [176]. Thereafter, this fungal strain has been used for the production of mycoprotein under the brand Quorn^TM^. In 1998, the fungal species used in Quorn^TM^ was re-classified as *Fusarium venenatum* based upon molecular phylogenetic, morphological and mycotoxin analyses [177]. More recently, genomic analysis described the difference between *F. graminearum* and *F. venenatum*, including genes that encode different types of mycotoxin (type B trichothecene in *F. graminearum* and type A in *F. venenatum*) [178]. To date, *F. venenatum* A3/5 is not known to produce mycotoxins in the processing conditions used, albeit regular monitoring is still used to ensure that the final product does not contain these toxins [176].

Typical composition of Quorn^TM^ mycoprotein includes protein (45%), fiber (25%), fat (13%) and carbohydrate (10%), as expressed per 100 g dry weight [179]. Manufacturing processes of Quorn^TM^ mycoprotein begin with aerobic fermentation of *F. venenatum* in a glucose-rich medium, along with other micronutrients, using an airlift bioreactor. After a desired level of recirculating solids is achieved, the fermenter broth is heated to stop growth and also to reduce the amount of RNA in the mycoprotein (approximately 1% *w/w*) through the action of natural nuclease enzymes in the mycelium. Centrifugation is then used to clarify the RNA-reduced fermenter broth, followed by vacuum chilling to achieve a final mycoprotein at approximately 24% (*w/w*) total solid contents [176,179,180]. Subsequent texturing can be done, for example, by the use of temperature (heat and chilled), pressure and in combination with egg proteins [180].

The main food safety hazard associated with mycoprotein is allergens. While data are limited, adverse reactions to mycoproteins have been reported in individuals with a history of mold allergies. In one report, a 15-year-old male exhibited type I hypersensitivity symptoms after consuming meatless chicken and subsequent skin prick test revealed that the individual was also allergic to several mold species, with particularly strong reaction to *Fusarium vasinfectum* [181]. Similarly, type I hypersensitivities were observed in a 27-year-old female within a few minutes of ingesting Quorn^TM^ burger, with detected IgE cross-reactivity with three mold species—namely, *Alternaria alternate*, *Aspergillus fumigatus* and *Cladosporium herbarm* [182]. This IgE-mediated allergy to mycoprotein may be caused by the acidic ribosomal protein P2, as previously determined in a 41-year-old male patient showing allergic reactions to Quorn^TM^ product [183]. In 2018, a group of researchers analyzed 2007 self-report adverse events related to Quorn^TM^ products from 1752 individual people and subsequently found that the majority of these events involved allergic reactions (hives and anaphylaxis) or gastrointestinal symptoms (vomiting and diarrhea) [184].

Future challenges in the mycoprotein research include finding sustainable carbon sources for long-term production, particularly to replace the highly refined glucose syrup currently in use. In addition to technological and industrial considerations, food safety must also be taken into account [176]. As previously highlighted, the production of mycotoxins varied with carbon sources [185,186], and thus these findings indicate a potential safety risk that may arise from the use of an alternative carbon source in mycoprotein production.

Agri-food wastes can potentially be utilized as a nutrient source in the production of mycoprotein. In one study, date wastes were used as a fermentation substrates for *F. venenatum*, and the authors found that the resulting mycoproteins did not result in allergic reactions in tested human subjects, that there was an absence of fumonisin gene expression in the starter culture and that no mycotoxins (zearalenone and deoxynivalenol) were detected in the fermentation medium, although low levels of lead (658 µg/kg), arsenic (161 µg/kg) and cadmium (30.57 µg/kg) were reported [187]. Future research can aim at assessing other waste materials for their potential use in mycoprotein production.

As summarized by Ritala et al., there are other fungal species considered for use as mycoprotein [144]. Currently, primary concerns related to mycoprotein (Quorn^TM^) are allergens, although mycotoxins may need to be considered when alternative carbon sources (other than high-refined glucose syrup) are used for growing different fungal species or heavy metals when the fermentation substrates are derived from agri-food wastes.

### 5.3. Bacterial Protein: Useful or Harmful Strains and RNA Contents

Bacterial protein contains about 50–80% protein on a dry weight basis [144]. Currently, research efforts on bacterial proteins are mostly aimed at their utilization as feed in farms or aquaculture, several of which are summarized in Table 4. Generally, bacteria can form biomass through autotrophy (utilizing carbon dioxide as a carbon source as mediated by light or chemical energy) or heterotrophy (non-carbon dioxide carbon source, for example, acetic acid, methanol, methane or formic acid) [188]. Bacterial biomasses can be grown in bioreactors using different growth substrates, including waste products from varying industries—for example, rhizospheric diazotrophs on brewery wastewater [189], *Methylococcus capsulatus* on methane gas [190] and *Rhodopseudomonas palustris* on latex rubber sheet wastewater enriched with pineapple extracts [191]. Among the available bacterial species, *M. capsulatus* has been found to promote health benefits in monogastric animals, such as minks, pigs and chickens [190], and thus future research on its potential as a protein source for humans is warranted. Feeding substrates supplemented with *M. capsulatus* also alleviated the severity of colitis and improved gut function in mice [192].

To avoid metabolic overload during the production of high-value products, co-cultures have been proposed, i.e., multi-species microbial consortia [197]. In one study, *Aspergillus niger* H3 and *Bacillus licheniformis* were used to produce bacterial protein from potato starch wastewater in a dual-step process: (1) *A. niger* H3 metabolized the fiber in the potato wastewater through the action of cellulases; (2) *B. licheniformis* utilized the released sugars from the fermentation of potato fibers to produce bacterial protein [198]. Others used a hybrid system of purple non-sulfur bacteria and aerobic heterotrophic bacteria to improve the production efficiency and nutritional quality of the bacterial protein produced, including higher protein content and more favorable amino acid/fatty acid profiles, as compared with when either of the bacteria was cultured alone [199].

Relative to other single-cell proteins (i.e., microalgal or mycoprotein), bacterial protein has the highest nucleic acid content at 15–16% [200]. During the metabolism of nucleic acid, purines (guanine and adenine) are degraded into uric acid. As humans lack the uricase enzyme, which is involved in the metabolism of uric acid in mammals, uric acid is usually excreted in the urine. However, impaired uric acid excretion leads to excessive accumulation of uric acid in the human body, which is a major cause of gout or hyperuricemia [201,202]. Thus, it is imperative that the nucleic acid content in bacterial proteins is reduced, for example, through heat treatments as used in the processing of mycoprotein.

Safety concerns can also arise from the use of toxin-producing bacterial species, such as *B. cereus* [203]. However, given the availability of non-pathogenic alternatives, for example, *Bacillus subtilis* [204], food manufacturers and researchers could circumvent the issue of toxins in bacterial proteins with relative ease. Nevertheless, precautions should still be taken, particularly when multispecies co-cultures are utilized. Another concern is the presence of potentially harmful substances within the cells used to produce bacterial proteins. For example, the bacterial strain *Cupriavidus necator* H16 contains non-nutritive polyhydroxybutyrate in their cytoplasm, which could accumulate in the organs upon ingestion, as previously reported for Sprague Dawley mice fed with the bacteria [195].

In addition, as wastewaters are commonly proposed as a growth substrate, surveillance systems are required to monitor the presence of biological and chemical hazards. As highlighted by Alloul et al., the cultivation of purple non-sulfur bacteria in wastewater was flanked with a variety of other bacteria, such as those belonging to the genera *Arcobacter*, *Dysgonomonas* and *Acinetobacter*, indicating potential quality control issues during the bacterial protein production [196].

## 6. Environmental Impact of Alternative Proteins

The environmental impact of cultured meat may vary depending on the growth medium used (Table 5). Regardless, current data suggest that the theoretical greenhouse gas emission, water use, eutrophication and land use in culturing meat are lower than conventional meat production, although cultured meat is still more energy intensive [28,205]. Similar to cultured meat, the environmental impact of insect-based food production depends on the type of feed used [206,207,208], with nutritious waste-based feeds being the most environmentally friendly [207].

Available life cycle analyses on two commercial plant-based meats—namely, Beyond Burger^®^ and Impossible Burger^®^—suggest that these products are more environmentally sustainable than conventional meats, as measured by their energy use, carbon emission, land use, water use and eutrophication. The environmental impact of these plant-based meats mainly occurs during the raw ingredient production [209,210].

Hydrogen-oxidizing autotrophic bacteria, such *C. necator*, can be turned into a sustainable protein source with a lower environmental impact than animal-based meat, including beef, fish and poultry [215]. As electricity consumption is the main driver of bacterial protein production, energy sources should be optimized for a mass-scale production [214,215]. Similarly, the cultivation of microalgae is the most energy intensive stage, and thus it contributes towards the majority of the environmental footprints associated with microalgal protein production [211], particularly when an open raceway pond is used [212].

Smetana et al. conducted a comparative study of different alternative proteins and found that cultured meat had the highest environmental impact (carbon emission and water use), as compared with mycoprotein and insect-based protein, including when caloric and protein contents were considered. However, the production of insect-based protein required the largest amount of land occupation, relative to mycoprotein and cultured meat [213]. Consistent with our data (Table 5), another study by Smetana et al. reported that microalgal proteins were more energy intensive, and thus had a higher carbon emission than other protein sources, including cultured meat, insect, yeast and bacteria [216]. Interestingly, the data collated in this review also indicate that cultured and plant-based meats have lower eutrophication potential than insect and single cell proteins (Table 5). To our knowledge, the two reports by Smetana et al. [213,216] are the only studies that directly compared the environmental impacts of these alternative proteins, and thus more research is required to establish a firm scientific framework for this issue. It is noteworthy that as functionalities of different food types can vary (for example, protein content or nutrient availability), direct comparison of the environmental impacts of different alternative proteins should be conducted with prudence.

## 7. Regulatory Framework

The food legislative framework is yet another necessary platform to ensure the safety of alternative proteins. Cultured meat, insect protein and single-cell protein are likely to be regulated as novel foods. EFSA is the key administrative institution for food safety in Europe and generally acts as expert consultants to the European Commission (EC). The latest EC novel food legislation Regulation (EC) 2015/2283 came into effect in 2018 and regulates the approval of foods derived from ingredients or production processes that were not used within the European Union prior to 15 May 1997 [217]—a guide document on the application process has also been published elsewhere [218]. Other relevant EC regulations include food hygiene regulations, such as Regulation (EC) 852/2003 and (EC) 853/2004 [219]. Alternative proteins can also be regulated through the Food, Drug and Cosmetic Act [220], including possible assessment for generally recognized as safe (GRAS) status [221].

In 2019, the Food and Drug Administration (FDA) and the United States Department of Agriculture’s Food Safety and Inspection Services (USDA-FSIS) announced a planned collaboration to regulate cultured meat. Under the agreement, FDA will oversee cell collection, cell banks, cell growth and cell differentiation, whereas USDA-FSIS will monitor post-harvest processes, including the production and labelling of the final cell-based food items [222]. Further, the Federal Meat Inspection Act and Poultry Product Inspection Act have also been proposed as supporting legislations [13]. In Europe, cultured meat may potentially be subjected to Regulations (EC) 1829/2003 and 1830/2003 [223,224], if GM cells are used (for example, iPSC).

Similarly, insect and bacterial proteins are subjects to the novel food legislation Regulation (EC) 2015/2283. In 2021, *T. molitor* was deemed safe for human consumption by the EFSA Panel on Nutrition, Novel Foods and Food Allergens, in compliance with the Regulations (EC) 2015/2283 [142]. Belluco, Halloran and Ricci summarized other supporting EC regulations relevant to edible insects [225]. Microalgal products must obtain GRAS status in the USA, for example, oil derived from *Ulkenia* sp. SAM2179, *Haematococcus pluvialis* extract containing astaxanthin esters, dried biomass of *A. platensis*, DHA-rich single-cell oil derived from *Crypthecodinium cohni*. In Europe, microalgal products that have historical human consumptions prior to 15 May 1997, such as *A. platensis*, can be approved according to the regular food safety standard Regulation (EC) 178/2002, but anything else is a subject to the novel food regulation [226].

In contrast, plant-based meats are regulated in a similar manner as other non-animal foods [227], particularly given that GM soybeans are deemed safe by the FDA [228] and EFSA [229]. Certain plant-based meats that contain soy leghemoglobin may be subjected to novel food regulation in Europe, although individual states can vary in their ways of regulating soy leghemoglobin in plant-based meats. In the USA, soy leghemoglobin has been declared as GRAS [86].

## 8. Research Gap and Future Outlook

Alternative proteins are a growing industry, and thus the global food sector should initiate collaborative efforts to ensure the safety of foods in this category. The main focus of these efforts should be to maintain food safety in a mass-scale production, including aspects related to allergens, pathogens, chemical contaminants and the environmental implications during production scale-ups. In this review, we have highlighted several potential safety risks associated with cultured meat, plant-based meat, insect protein and single-cell protein.

There is a lack of research on the safety of cultured meat, with most studies focusing on technological improvements for better production means. Infectious prion and viruses are potentially the main hazards related to cultured meat production using serum-based media. Thus, future developments of methods for removing these contaminants are warranted, such as the use of hollow fiber anion-exchange membrane chromatography to remove prion from large volumes of cell culture media [230]. Concerns about the introduction of foreign genes, such as during the conversion of somatic cells into iPSC, may be circumvented by the use of small molecules as an alternative cell reprogramming system [231]. In the current literature, it appears that antibiotics are not used in the production of cultured meat, primarily based upon the notion that this alternative protein is produced in a highly controlled and closely monitored environment [13,232]. However, to our knowledge, there have not been any studies addressing this issue using verifiable data, and thus we encourage the scientific community to investigate this issue further, including through the provision of assessments of the safety measures used to control biological contaminants in cultured meat without antibiotics.

Available data suggest that plant-based meat may contain allergens, anti-nutrients or traces of glyphosate, although activities of these compounds may be reduced by heat treatments. In the future, there is also a need for discussion of the health implications of extensive processing (i.e., ultraprocessed) involved in the production of plant-based meat, including potential development of carcinogens during the thermal treatments.

Allergens are one of the primary safety issues associated with the consumption of insects, but the clinical significance of this is yet to be established. Future research can aim at identifying the types of allergen present in different edible insect species, and subsequently assessing their health effects across demographics, i.e., by age, allergy status, ethnicity, etc. Microbiological content of insects also varies with species, and future mass-scale production of edible insects would require careful selection of those species harboring bacteria communities that are less pathogenic to humans.

Toxins pose a health risk related to single-cell protein. In microalgae, this is primarily due to environmental cross-contamination, which indicates the importance of choosing appropriate cultivation reservoirs. For mycoprotein, allergens are the main hazard, and future research is necessary to identify the risk factors associated with mycoprotein allergies. When bacteria are used as single-cell protein, careful selection of non-pathogenic bacterial strains is paramount.

In the current literature, the regulatory framework for novel foods has been described based upon food standards in Europe and the USA. As alternative proteins are a global strategy to mitigate climate and environmental issues, the scientific community should expand the scope of the discussion to include food standards in other parts of the world. For example, Australia-New Zealand Food Standard Code Standard 1.5.1 describes the pre-market assessment criteria for novel foods intended for sale in Australia and New Zealand [233], or Schedule 25 lists the approved novel food products, including several that are derived from microalgae [234]. Soy leghemoglobin has also been approved in these two countries [235].

## Figures and Tables

**Figure 1 foods-10-01226-f001:**
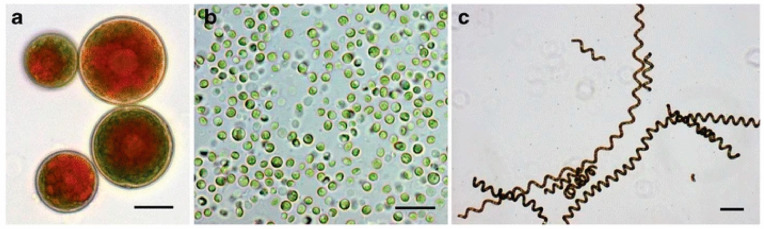
Microalgae that can be used as sources of protein for human consumption: (**a**) *Haemotococcus pluvialis* with droplets of astaxanthin within the cells; (**b**) *Chlorella vulgaris*; (**c**) *Arthrospira maxima* SAG 21–99 (spirulina). Scale bar = 15 µm. Images were taken from Wells et al. (2016) published under the Creative Commons Attribution 4.0 International License (http://creativecommons.org/licenses/by/4.0/) [152].

**Table 1 foods-10-01226-t001:** Relevant technologies to future developments of cultured meat.

Technology	Relevance	Main Finding	Reference
Bovine embryonic stem cells	Embryonic stem cells that are derived from livestock animals can be transformed into any cell type.	First report on the derivation of stable bovine embryonic stem cells in a culture containing fibroblast growth factor 2 and an inhibitor of Wnt signaling pathways.	[18]
Pluripotent ^1^ stem cells derived from adult fibroblasts (iPSC)	Ethical issues on the use of embryonic stem cells may be circumvented.	Induction of pluripotent stem cells from mouse embryonic or adult fibroblasts by the introduction of four transcription factors (OCT3/4, Sox2, c-Myc and Klf4). Another transcription factor commonly associated with pluripotency, namely NANOG, was dispensable.	[19]
Skeletal muscles derived from porcine iPSC	Generation of livestock tissues from iPSC.	Contractile porcine myotubes were produced through a coordinated application of CHIR 99021 (inhibits GSK3B enzyme), 5-aza-cytidine (DNA methylation inhibitor) and ectotopically expressed MYOD1.	[20]
Co-culture of IMP and MSC derived from chicken	Co-culture system is required to produce a complex tissue resembling conventional meat.	IMP and MSC were successfully co-cultured using a transwell chamber. In proliferative stage, MSCs accelerated the differentiation of IMPs, which resulted in a higher fat content in co-cultured IMPs than single-cultured ones. Opposite effect was observed in non-proliferative stage.	[21]
Isolation of bovine PA and its adipogenic differentiation	Method for culturing adipocytes in vitro allows for future development of cultured meat that contains fat components.	Descriptions of protocols for isolating pre-adipocyte (multipotent stem cell) from primary bovine adipose tissue and for their subsequent differentiation in 2D culture media or on 3D alginate scaffolds. Plant- and animal-based free fatty acids used in the differentiation process.	[22]

^1^ Pluripotent stem cells refer to those that can be transformed into any type of cell, as opposed to multipotent stem cells that can only be differentiated into specific cell types. iPSC, induced pluripotent stem cells; GSK3B, glycogen synthase kinase 3-β; MSC, muscle satellite cells; IMP, intramuscular pre-adipocyte; PA, pre-adipocyte.

**Table 2 foods-10-01226-t002:** Ingredients of plant-based meat as proposed by Egbert and Borders in 2006 [47].

Ingredient	Function	Usage Level (%)
Water	Distribution of ingredients, emulsification, juiciness	50–80
TVP = textured soy flour, textured soy concentrate, textured wheat gluten or textured protein combinations (for example, soy and wheat)	Water binding, texture/mouthfeel, appearance, protein fortification/nutrition and source of insoluble fiber	10–25
Non-texture proteins = ISP, functional soy concentrate, wheat gluten, egg whites * or whey proteins *	Water binding, emulsification, texture/mouthfeel and protein fortification/nutrition	4–20
Flavors/spices	Flavor (savory, meaty, roasted, fatty and serumy), flavor enhancement (for example, salt) and mask cereal notes	3–10
Fat/oil	Flavor, texture/mouthfeel, succulence and Maillard reaction/browning	0–15
Binding agents = wheat gluten, egg whites *, gums and hydrocolloids, enzymes or starches	Texture, water binding, potential fiber content and determine processing conditions (depending on how and where they are added)	1–5
Coloring agents = caramel colors, malt extracts, beet powder and other FDA-approved colors (FD & C)	Appearance and eye appeal	0–0.5

TVP, texture vegetable proteins; ISP, isolated soy proteins; FDA, Food and Drug Administration. * These ingredients are not plant-based and their use in plant-based meat products requires clear labelling.

**Table 3 foods-10-01226-t003:** Major groups of insects consumed around the world as reported by van Huis et al. in association with the Food and Agriculture Organization of the United Nations (FAO) [88]. Several species from each class are mentioned, but this list is not exhaustive.

Insect Class	Insect Species	Percentage (%) ^2^
Coleoptera ^1^	Yellow mealworm (*Tenebrio molitor*), palm weevil (*Rhynchophorus phoenicis, Rhynchophorus ferrugineus* and *Rhynchophorus palmarum*)	31
Lepidoptera ^1^	Caterpillars of butterflies or moths (*Daphnis* spp., *Theretra, Imbrasia belina, Omphisa fuscidentalis*, *Comadia redtenbacheri or Aegiale hesperiasis*)	18
Hymenoptera ^1^	Weaver ants (*Oecophylla* spp.), yellow jacket wasps (*Vespula* and *Dolichovespula* spp.) and honeybees (*Apis mellifera*)	14
Orthoptera	Crickets (*Gryllus bimaculatus* and *Acheta domesticus*) and locusts (*Locusta migratoria*)	13
Hemiptera (suborders Homoptera and Heteroptera)	Cicadas (*Ioba, Playtypleura* and *Pycna*), pentatomid bugs (*Agonoscelis versicolor*)	10
Isoptera, Odonata, Diptera and others	Termites (*Macrotermes* and *Syntermes*), dragonflies, flies and others	14

^1^ Edible insects listed are usually consumed during their larval stage. ^2^ Proportion of edible insects from each class, with the total reported number of edible insects coming from 1900 species.

**Table 4 foods-10-01226-t004:** Several in vivo studies of bacterial protein fed to live animals.

Bacterial Species	Live Animal	Main Finding	Reference
*Methylococcus capsulatus* (Bath)	Female C57BL/6NTac mice	Mice fed with the bacteria exhibited less profound colitis symptoms and higher colonic epithelial layer (increased cell proliferation and mucin 2 transcription) than those in the control groups.	[192]
*M. capsulatus* (Bath)	Japanese yellow tail fish (*Seriola quinqueradiata*)	Growth rate and feeding efficiency of fish fed with bacterial protein was the same as those fed with conventional fish meal but only up to a bacterial protein concentration of 20%, beyond which both parameters were negatively affected.	[193]
*Methylobacterium extorquens*	Rainbow trout (*Oncorchynchun mykiss* Walbaum)	Fish fed with bacterial protein (5% or 10%) had similar feeding efficiency to the control groups, with survival improved for fish in the 10% bacterial protein group.	[194]
*Cupriavidus necator* H16	Sprague Dawley mice	Mice fed with the bacteria (experimental group) had lower weight gain over 28 days of feeding trial than those in the control groups, with PHB detected in the excrements of the experimental mice. Kidneys, ileums and stomachs of the experimental mice were also heavier, potentially due to the accumulation of PHB in the murine organs.	[195]
*Rhodobacter capsulatus* or *Rhodopseudomonas palustris*	White leg shrimp (*Penaeus vannamei*)	Shrimps fed with either of two bacterial species had higher feed conversion rates and individual weights than those in the control group. Tolerance of ammonia was also higher in shrimps fed with *R. palustris*, relative to the control group.	[196]

PHB, polyhydroxybutyrate.

**Table 5 foods-10-01226-t005:** Life cycle analyses (LCA) of alternative proteins.

Protein Type	Energy Use (MJ/kg)	GHG Emission (kg CO_2_-eq/kg Product)	Water Use or Eutrophication ^a^	Land Use (m^2^a/kg) ^b^	Reference
**Cultured meat**					
Minced beef ^1^	26–33	1.90–2.24	0.36–0.52 m^3^/kg meat (W)	0.19–0.23	[28]
CHO ^2^	106	7.5	7.9 g PO_4_-eq/kg meat (E)	5.5	[205]
**Plant-based meat**					
Beyond Burger^®^	54.15	3.35	28.84 m^3^/kg meat (W)	3.97	[209]
Impossible Burger^®^	NA	3.5	0.11 m^3^/kg meat (W); 1.3 g PO_4_-eq/kg meat	2.5	[210]
**Insect protein**					
Mealworm (*T. molitor* and *Zophobas morio*)	33.68	2.65	NA	3.56	[206]
Black soldier fly (*H. illucens*)	21.20–99.60	1.36–15.10	NA	0.032–7.03	[207]
Cricket (*G. bimaculatans* and *A. domesticus*)	NA	2.29	0.43 m^3^/kg cricket (W); 0.00047 kg P-eq and 0.020 kg N-eq/kg cricket (E)	NA	[208]
**Single-cell protein**					
Spirulina tablets (*A. platensis*)	7.88–12.7	5.05–7.71	0.015–0.022 kg N-eq/kg tablet (E)	NA	[211]
Micoalgal protein (*A. platensis*)	1225.6–3338.3	78.1–196.3	3.2–3.3 m^3^/kg protein meal (W); 49.2–85.3 kg N-eq/kg protein meal (E)	1.7–4.3	[212]
Microalgal protein (*C. vulgaris*)	217.1–4181.3	14.7–245.1	0.3–3.9 m^3^/kg protein meal (W); 40.6–105.3 kg N-eq/kg protein meal (E)	1.9–5.4	[212]
Mycoprotein	60.07–76.8	5.55–6.15	NA	0.79–0.84	[213]
Bacterial protein (*Cupriavidus necator*)	NA	0.81–1	0.0001–0.0038 m^3^/kg protein (W); 0.000333 kg P-eq/kg protein (E)	0.029–0.085	[214]
Bacterial protein (hydrogen-oxidizing bacteria)	200	8	2.5 m^3^/kg protein (W); 0.0025 kg P-eq/kg protein and 0.00035 N-eq/kg protein (E)	0.8	[215]

GHG, greenhouse gas; NA, not available. ^a^ Water use (W) is expressed in volumetric unit (m^3^ or L/weight of product), whereas eutrophication (E) is a measure of the amount of contaminants released into freshwater or marine environments (g contaminant/weight of product). ^b^ Land use is expressed in annual area occupation (m^2^a). ^1^ Cyanobacterial hydrolysate was assumed as the growth medium. ^2^ LCA was conducted based on the proliferation of Chinese hamster ovary (CHO) in a growth medium mainly comprising basal medium and soy hydrolysate.

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
