# Peer review of "Safety of Alternative Proteins: Technological, Environmental and Regulatory Aspects of Cultured Meat, Plant-Based Meat, Insect Protein and Single-Cell Protein"

_foods, 2021, doi:10.3390/foods10061226_

Round 1

Reviewer 1 Report

Very clearly, and concisely processed overview of the topic, with specific data.

Authors do not list generally known information in the field, which is exceptional and pleasant.

The chosen topic is highly comprehensive, which inevitably leads to simplifying all issues. Still, this review can be very well used as an introduction to the safety of alternative proteins.

Reviewer 2 Report

General comments

The review paper “Safety of Alternative Proteins: Technological, Environmental and Regulatory Aspects of Cultured Meat, Plant-based Meat, Insect Protein and Single-Cell Protein” describes in a comprehensive way the state of the art of research in these interesting and innovative domains.

Due to the huge amount of literature considered a more systematic approach in the search and selection of relevant literature would have made this review more grounded from a methodological point of view. However, I acknowledge that the breadth of investigated topic would have made the effort very demanding.

The paper is well structured and it is easy to find information, please check paragraph numbering as there are some mistake (see for example 3.1 used twice).

I appreciate the comment in line 784-792 about the lack of food safety studies, Often they are under considered during the study of this novel foods but at the end their lack impair the evaluation process by food safety authorities. Maybe you could stress this concept more, even in the abstract.

Specific comments

Line

Comment

151

Pay attention to the “lack of data” assertion as you have not used a systematic approach to find relevant literature, may yo have missed something?

163

Whey, thus not totally plant based (vegan) or animal free companies (line 167).

214

3.1 should be 3.2

349

Metagenetic? Maybe should e metagenomics or metatassonomic

348-358

I do not understand the food safety implications (if any) of this paragraph.

634

6.2 should be 6.3

719-723

Are you sure about this statement about land occupation? I do not find this info in Smetana et al. Please be careful and acknowledge the fact that this consideration is based on a single study…

739-740

Check this statement. Spirulina (a platense) is not a novel food.

Reviewer 3 Report

Dear authors

I carefully read the manuscript proposed regading the new "sources of foods"

The manuscript is very well structured and described point by point the potential safety issues of these new materials. Also, authors presented a large and current bibliography review.

The only point not assessed is the use of antibiotics in celular growth to obtain cultured meat. Does it represent a food safety or public health hazard?.

The manuscript presented a great quality. If possible, some information regarding the use of antibiotics in cultured cells may be added to the text.
